# Comparing Liver Venous Deprivation and Portal Vein Embolization for Perihilar Cholangiocarcinoma: Is It Time to Shift the Focus to Hepatic Functional Reserve Rather than Hypertrophy?

**DOI:** 10.3390/cancers15174363

**Published:** 2023-09-01

**Authors:** Rebecca Marino, Francesca Ratti, Angelo Della Corte, Domenico Santangelo, Lucrezia Clocchiatti, Carla Canevari, Patrizia Magnani, Federica Pedica, Andrea Casadei-Gardini, Francesco De Cobelli, Luca Aldrighetti

**Affiliations:** 1Hepatobiliary Surgery Division, IRCCS San Raffaele Hospital, 20132 Milan, Italy; marino.rebecca@hsr.it (R.M.); ratti.francesca@hsr.it (F.R.); clocchiatti.lucrezia@hsr.it (L.C.); 2Department of Radiology, IRCCS San Raffaele Hospital, 20132 Milan, Italy; dellacorte.angelo@hsr.it (A.D.C.); santagelo.domenico@hsr.it (D.S.); decobelli.francesco@hsr.it (F.D.C.); 3Experimental Imaging Center, IRCCS San Raffaele Scientific Institute, 20132 Milan, Italy; 4Nuclear Medicine Department, San Raffaele University and Research Hospital, 20132 Milan, Italy; canevari.carla@hsr.it (C.C.); magnani.patrizia@hsr.it (P.M.); 5Pathology Unit, Department of Experimental Oncology, San Raffaele Hospital, 20132 Milan, Italy; pedica.federica@hsr.it; 6Department of Medical Oncology, IRCCS San Raffaele Hospital, 20132 Milan, Italy; casadeigardini.andrea@hsr.it; 7Faculty of Medicine, University Vita-Salute San Raffaele, 20132 Milan, Italy

**Keywords:** PeriHilar cholangiocarcinoma, liver hypertrophy techniques, portal vein embolization, liver venous deprivation, functional outcomes, volumetric outcomes, liver surgery

## Abstract

**Simple Summary:**

Liver venous deprivation (LVD) has emerged as a promising technique in the pursuit of improving surgical outcomes for perihilar cholangiocarcinoma (PHC) patients. This procedure, which combines portal inflow and hepatic outflow abrogation, has generated significant clinical interest. However, its specific role in optimizing the future liver remnant (FLR) before liver resection, especially when compared to portal vein embolization (PVE), remains unclear. Between 2013 and 2022, all patients with PHC undergoing preoperative FLR enhancement were evaluated. FLR volume assessments were conducted at two time points to evaluate early and late efficacy indicators. While both LVD and PVE cohorts experienced similar post-procedural complications, LVD demonstrated superior FLR function and growth rates at both assessment points. This suggests faster recovery and improved remnant liver functionality. Although FLR volumes remained comparable between the techniques, LVD emerged as an effective method for optimizing FLR in PHC, potentially enhancing liver function and reducing post-hepatectomy liver failure rates, thus improving overall surgical outcomes.

**Abstract:**

**Purpose**: Among liver hypertrophy technics, liver venous deprivation (LVD) has been recently introduced as an effective procedure to combine simultaneous portal inflow and hepatic outflow abrogation, raising growing clinical interest. The aim of this study is to investigate the role of LVD for preoperative optimization of future liver remnant (FLR) in perihilar cholangiocarcinoma (PHC), especially when compared with portal vein embolization (PVE). **Methods**: Between January 2013 and July 2022, all patients diagnosed with PHC and scheduled for preoperative optimization of FTR, through radiological hypertrophy techniques, prior to liver resection, were included. FTR volumetric assessment was evaluated at two distinct timepoints to track the progression of both early (T_1_, 10 days post-procedural) and late (T_2_, 21 days post-procedural) efficacy indicators. Post-procedural outcomes, including functional and volumetric analyses, were compared between the LVD and the PVE cohorts. **Results**: A total of 12 patients underwent LVD while 19 underwent PVE. No significant differences in either post-procedural or post-operative complications were found. Post-procedural FLR function, calculated with (99m) Tc-Mebrofenin hepatobiliary scintigraphy, and kinetic growth rate, at both timepoints, were greater in the LVD cohort (3.12 ± 0.55%/min/m^2^ vs. 2.46 ± 0.64%/min/m^2^, *p* = 0.041; 27.32 ± 16.86%/week (T_1_) vs. 15.71 ± 9.82%/week (T_1_) *p* < 0.001; 17.19 ± 9.88%/week (T_2_) vs. 9.89 ± 14.62%/week (T_2_) *p* = 0.034) when compared with the PVE cohort. Post-procedural FTR volumes were similar for both hypertrophy techniques. **Conclusions**: LVD is an effective procedure to effectively optimize FLR before liver resection for PHC. The faster growth rate combined with the improved FLR function, when compared to PVE alone, could maximize surgical outcomes by lowering post-hepatectomy liver failure rates.

## 1. Introduction

To this day, extended liver resection leading to R0 resection margins is the sole viable choice offering the potential for long-term survival in patients with Perihilar Cholangiocarcinoma (PHC) [1,2]. Due to standardized surgical approach for hepatic hilum tumors, which mandates (extended) right or left hepatectomies in conjunction with bile duct and caudate lobe resection in Bismuth III or IV cases, postoperative morbidity and mortality rates rank among the highest within the HPB field [3,4]. Most patients affected by PHC and scheduled for extensive parenchymal resections face increased perioperative risks linked to disease location, jaundice, preoperative cholangitis potentially leading to sepsis, and impaired remnant liver function, heightening postoperative liver failure risk and creating the perfect thunderstorm potentially jeopardizing surgical effort. The primary cause of increased postoperative mortality remains indeed post-hepatectomy liver failure (PHLF), while the most prevalent factor limiting the achievement of negative margin resections (R0) is inadequate future liver remnant (FLR) [5,6,7] Strategic implementation of preoperative optimization protocols–including evaluation of FLR and ruling out the need for hypertrophy techniques-is crucial for accurate preparation and selection of suitable surgical candidates. Liver hypertrophy techniques play a central role in inducing adequate liver regeneration, thereby ensuring the attainment of permissible FLR volumes to control PHLF risk [8].

Portal vein embolization (PVE) is presently acknowledged as the gold standard hypertrophy technique in PHC patients, linked to reduced rates of PHLF and mortality, alongside maintaining minimal post-procedural complications (2.2–3.1%) [9]. Although PVE has generally exhibited effective contralateral hypertrophy induction within 4–6 weeks, approximately 20% of treated patients are excluded from surgical resection due to the emergence of absolute contraindications, such as inadequate liver regeneration or tumor progression [10]. To address the limitations of PVE and accelerate FLR growth while improving its function, two alternative strategies have been suggested: the combination of liver partition and portal vein ligation for staged hepatectomy (ALPPS) [11] and liver venous deprivation (LVD) [12]. The ALPPS procedure has demonstrated the ability to rapidly induce liver hypertrophy, allowing for resection within 7 to 10 days. However, it requires an additional surgical intervention, leading to a significantly elevated 90-day mortality rate of 20% to 48%, particularly in cases of PHC where it currently finds no clinical indication [13]. On the contrary, LVD (simultaneous endovascular occlusion of both portal and hepatic veins) is a recently described technique which has shown a notable reduction in drop-out rates from surgical resection and higher degrees of hypertrophy (DH) compared to standalone PVE, all while maintaining an equivalent safety profile [14]. These favorable results have been documented in cases of secondary liver tumors and HCC. Nevertheless, there is currently a gap in our understanding of the application of LVD in the context of PHC. This area merits particular attention, especially considering the high-risk nature of these patients, both in terms of the perioperative and oncological outcomes. Given the promising results of LVD and the absence of available series specifically addressing its application in the preoperative management of PHC patients scheduled for major hepatectomies, the aim of this study is to assess the efficacy and the safety of LVD in PHC, by comparing FLR hypertrophy and functional outcomes with standalone PVE.

## 2. Material and Methods

### 2.1. Study Population 

Between January 2013 and July 2022, data were retrieved and retrospectively analyzed from the prospectively maintained database of the Hepatobiliary Surgery Unit of San Raffaele Hospital in Milan for all consecutive patients diagnosed with PHC. All patients with a diagnosis of PHC from 2013 on are indeed preoperatively treated according to the institutional perioperative optimization protocol, including a specific evaluation of FLR volume and function to assess the indication for hypertrophy techniques prior to surgical resection. From January 2013 to June 2019, PVE was the preferred hypertrophy technique for all PHC cases, while starting from July 2019, LVD was introduced into the clinical practice to fully replace PVE. To assess the safety and efficacy profile of LVD, volumetric, functional, and postoperative outcomes were compared between the LVD cohort (July 2019–July 2022) and the PVE cohort (January 2013–June 2019). The study design is provided in Figure 1. 

### 2.2. Preoperative Workup 

The standard preoperative assessment has been described elsewhere [15,16]. Computed tomography and magnetic resonance cholangiopancreatography were conducted for all patients. Indication to perform right hepatectomy/trisectionectomy was based on both longitudinal and radial disease extension leading to biliary and/or vascular infiltration. Volumetric and functional evaluation through 99-mTc mebrofenin hepatobiliary scintigraphy (HBS) is described in detail later. 

In cases necessitating the adoption of liver hypertrophy techniques due to insufficient future liver remnant (FLR) volume or function, a percutaneous transhepatic biliary drainage (PTBD) was positioned in the FLR. Biliary decompression, achieved by placing a PTBD in the FLR was standardly performed in candidates with hypertrophy techniques. Eventually, biliary stenting of the right bile ducti, via endoscopic retrograde cholangiography, was indicated based on the total bilirubin level following PTBD. The therapeutic approach and ultimate decision regarding surgical resection for each case were routinely discussed during weekly multidisciplinary PHC board meetings. This meeting includes the HPB surgical team, oncologists, diagnostic and interventional radiologists, pathologists, HPB endoscopists, and the navigator nurse responsible for the patient’s care.

### 2.3. Inclusion Criteria for Liver Hypertrophy

A quadriphasic computed tomography scan or magnetic resonance imaging with gadoxetic acid was conducted within a 30-day period prior to the procedure. Liver volume was computed using specialized software (Philips Intellispace Portal, Version 12.1; Philips, Amsterdam, The Netherlands), excluding tumors and major vessels to accurately assess the functional volume. Following this, segmentation was performed based on the Couinaud classification. The calculation of the future liver remnant (FLR) was based on the functional volume and was determined in alignment with the specific surgical requirements. The Vauthey formula was used to define the standardized liver volume (sTLV) [17]. The standardized FLR ratio (sFLR) was defined as the ratio between FLR volume and standardized liver volume (Figure 2a,b).

Hepatobiliary scintigraphy (HBS) was conducted before the procedure, following established protocols [18] The calculation of the future liver remnant function (FLRF) involved delineating the counts within the FLR, dividing this by the total liver counts, and then multiplying this factor with the total liver (99m) Tc-Mebrofenin uptake. The result was expressed as a percentage per minute per square meter (%/min/m^2^). (Figure 2c,d) [19].

Patients were considered eligible for liver hypertrophy technique (PVE or LVD) before hepatectomy based on the following criteria: disease affecting the right lobe ± segment IV, and, most importantly, FLR volume less than 35% (in patients with prior cholestasis and/or liver steatosis)), or less than 40% in cases of cirrhosis, and/or FLR function measured by (99m) Tc-Mebrofenin HBS indicating less than 2.69%/min/m^2^. If volumetric and/or functional cutoffs were not reached during the evaluation at diagnosis, it is an indication that hypertrophy techniques were given. 

### 2.4. Portal Vein Embolization and Hepatic Vein Embolization Technique

Using ultrasound guidance, a 21-Gauge Chiba introducer needle was directed towards a branch of the right portal vein. Subsequently, a stepwise process involving the insertion of a 0.018-inch Cope guidewire (Boston Scientific, Natick, MA, USA) followed by a triaxial 4.5 Fr introducer system (Accustick II, Boston Scientific) was undertaken, leading to portography. The embolization procedure utilized a coaxial setup consisting of a 4.5 Fr introducer, a 4 Fr catheter, and a 2.7 Fr microcatheter (Renegade high flow, Boston Scientific). After flushing the microcatheter with a 5% glucose solution, a combination of cyanoacrylate (Glubran II; GEM, Viareggio, Italy) and iodized oil (Lipiodol Guerbet, Aulnay-sous-Bois, France) in a ratio of 1:5 or 1:6 was injected for embolization. To ensure closure of the access route, distal occlusion was accomplished using the same cyanoacrylate-oil mixture while retracting the introducer.

Immediately after PVE, the decision to embolize the right hepatic vein was made based on the operator’s discretion and the feasibility of percutaneous access through healthy liver tissue. This embolization of the right hepatic vein was achieved either via a transhepatic or transjugular approach.

Utilizing the transhepatic approach, the right hepatic vein was accessed under the guidance of ultrasound using a 22-Gauge Chiba introducer needle. Following this, a 0.018-inch Cope guidewire and a 4.5 Fr triaxial system were introduced, leading to the performance of phlebography. Subsequently, a 7 Fr transhepatic introducer sheath was advanced over a 0.035 guidewire. An Amplatzer Plug II occlusion device (Abbott) was deployed, with its diameter selected to allow for a 50–100% oversizing relative to the vein. The distal marker of the occlusion device was positioned 20-to-30 mm from the confluence of the hepatic vein in the inferior vena cava. To achieve occlusion, a 1:5 or 1:6 mixture of cyanoacrylate and oil was utilized, and applied distal to the route of the occlusion device during the retraction of the introducer.

In the transjugular approach, the right internal jugular vein was accessed using the same method. Similarly, a 7 Fr introducer sheath was positioned in the hepatic vein, and an Amplatzer Plug II occlusion device (Abbott) was then deployed, as previously described (Figure 3a–c).

### 2.5. Post-Procedural Evaluation

In accordance with institutional guidelines, a CT scan involving volumetric reassessment of the future liver remnant (FLR) was conducted at two specific time points. The first assessment (T_1_) was ideally performed around 10 days after the procedure to capture early efficacy, while the second assessment (T_2_) was ideally conducted after 21 days. The CIRSE Classification system was used to grade post-procedural complications. 

Concerning hypertrophy parameters, the degree of hypertrophy (DH) was defined as the percentage difference between FLR (%) at a given time point and the baseline FLR (%). 

The calculation for FLR increase was as follows: FLR increase = (FLR_post-procedural_ − FLR_baseline_) × 100%

The kinetic growth rate (KGR) was calculated according to the following formula: KGR = DH(%)/time elapsed since hypertrophy (weeks). HBS was ultimately conducted to validate the surgical viability (threshold > 2.69%/min/m^2^).

### 2.6. Statistical Analysis 

The statistical analyses were performed with IBM SPSS Statistics 28 (IBM Corp., Armonk, NY, USA). The Shapiro-Wilk normality test was conducted to evaluate the normality of the distribution. Continuous variables exhibiting normal distribution were presented as means ± standard deviation, while those with non-normal distribution were expressed as medians with their respective ranges. To analyze continuous variables, the Student’s *t*-test was employed for normally distributed data, and the Mann-Whitney U test for independent samples was used for non-normally distributed data. The data was subjected to descriptive assessment and frequencies were employed for categorical or ordinal variables. Qualitative variables were assessed using either the χ^2^ test or Fisher’s exact test, as deemed suitable. A significance level of *p* < 0.05 was adopted to determine statistical significance.

## 3. Results

### 3.1. Baseline Characteristics

During the study timeframe, 12 patients underwent LVD and 19 patients underwent PVE. Baseline characteristics are reported in Table 1. Age distribution was similar between the groups, with LVD patients having a mean age of 68.61 ± 8.06 years and PVE patients having a mean age of 66.53 ± 8.22 years (*p* = 0.494). Gender distribution was also balanced, as 58.3% of the LVD group were male, compared to 52.6% in the PVE group (*p* = 0.867). Notably, a majority of cases in both groups were characterized by Bismuth Type III and IV tumors, constituting 91.7% and 8.3% respectively in LVD, and 89.5% and 10.5% respectively in PVE (*p* = 1.000). Baseline parameters including BSA, ASA score, preoperative biliary drainage, preoperative cholangitis, total bilirubin, AST, ALT, and presence of liver cirrhosis were comparable between the two cohorts. In terms of functional assessment, FLR function displayed no significant difference between LVD (1.92 ± 0.14%/min/m^2^) and PVE (1.87 ± 0.62%/min/m^2^) indicating similar baseline functional liver capacity (*p* = 0.854). Similarly, no differences were found between the two cohorts in the pre-procedural volumetric analysis.

### 3.2. Post-Procedural Volumetric and Functional Evaluation 

A comparative analysis of volumetric outcomes between the LVD group and the PVE group at two distinct post-procedural timelines (T_1_ and T_2_) is presented in Table 2a,b. The first volumetric assessment (12.4 ± 4 days post-procedural), aimed at capturing early efficacy indicators, revealed no significant differences in FLR volume between the LVD and PVE groups. However, the LVD group exhibited a significantly greater differential (PostFLR–PreFLR) volume change (190.50 ± 116.50 mL vs. 112.28 ± 84.74 mL, *p* = 0.034), indicating more substantial early volume augmentation. Furthermore, the LVD group demonstrated a significantly higher sDH (48.45 ± 26.87% vs. 39.67 ± 22.38%, *p* = 0.006) and kinetic growth rate per week (KGR/week) (27.32 ± 16.86% vs. 15.71 ± 9.82%, *p* < 0.001). At T_2_ (21.3 ± 4 days post-procedural), representing the second volumetric assessment, again no significant differences were observed in FLR volume. Nevertheless, the LVD group showed a significantly larger differential (PostFLR–PreFLR) change (216.38 ± 118.43 mL vs. 125.41 ± 93.45 mL, *p* = 0.036), underscoring sustained volume augmentation. Similarly, sDH remained significantly higher in the LVD group (55.60 ± 31.02% vs. 42.29 ± 21.09%, *p* = 0.003), and the LVD group maintained a significantly higher KGR/week rate (17.19 ± 9.88% vs. 9.89 ± 14.62%, *p* = 0.034). Other parameters, including cFLR and sFLR, displayed no significant differences between the groups at both assessment timelines.

The post-procedural functional analysis, shown in Table 2c, revealed that the availability of pre- and post-procedural scintigraphy was comparable between the LVD group and the PVE group, with 83.3% and 84.2% availability, respectively (*p* = 0.806). Regarding postprocedural FLR function, the LVD group exhibited a significantly improved liver function compared to the PVE group (3.22 ± 0.55%/min/m^2^ vs. 2.62 ± 0.64%/min/m^2^; *p* = 0.041).

### 3.3. FLR Volume Gain 

The percentage of FLR volume gains from baseline within the LVD and PVE cohorts across the two study timelines, T_1_ and T_2_, are shown in Figure 4. The mean percentage of FLR volume gains is delineated by the blue line for LVD and the orange line for PVE, accompanied by their respective standard deviations (SD).

At T_1_, in comparison to the baseline, the PVE group exhibited a FLR volume gain of 31.2% (SD ± 11.2), while the LVD group demonstrated a more pronounced enhancement of 48.1% (SD ± 9.7).

Moving to T_2_, the trends persisted, with the PVE group showcasing a 42.8% FLR volume gain (SD ± 12.8), and the LVD group presenting a significant 52.5% expansion (SD ± 15.3). Importantly, the disparities in percentage FLR volume gains between the LVD and PVE groups were statistically significant (*p* < 0.001) at both T_1_ and T_2_.

### 3.4. FLR Functional Gain

A scatter plot was constructed to visually represent the distribution of baseline and postprocedural FLR functional values (Figure 5). Baseline values were depicted in blue, while postprocedural LVD values were shown in green. Notably, the LVD group demonstrated a remarkable improvement in liver function. Their baseline FLR function showed a substantial percentage increase of approximately 67.71%, reaching a postprocedural value of 3.22 ± 0.55%/min/m^2^. In comparison, the PVE group displayed an increase of approximately 40.16% in FLR function, resulting in a postprocedural value of 2.62 ± 0.64%/min/m^2^. These differences were statistically significant, with the mean and standard deviation of the LVD group surpassing those of the PVE group (*p* = 0.041). The scatter plot effectively contrasts the baseline and postprocedural functional values, visually illustrating the LVD group’s significant improvement compared to their baseline values.

### 3.5. Post-Procedural Outcomes

The evaluation of post-procedural outcomes is reported in Table 3. 

Two patients, one from each group, experienced post-procedural complications. In both cases, these complications manifested as segmental portal thrombosis in segment 2, as observed at the first post-procedural assessment (T_1_). However, both cases were successfully treated conservatively using therapeutic doses of low molecular weight heparin (grade-2). No additional procedural complications, such as bilomas, hepatic bleeding, or arteriovenous fistulas (AVF) were observed.

The time to surgery was comparable between the groups (29 ± 3 days for LVD and 35 ± 4 days for PVE, *p* = 0.368). There were no cases of post-hepatectomy liver failure (PHLF, ISGLS B/C) in the LVD group, while one case (5.26%) occurred in the PVE group (*p* = 0.613). Surgery drop-out reasons included oncological progression for 2 (10.5%) patients after PVE, while no drop-outs occurred in the LVD group (*p* = 0.509). No instances of insufficient future liver remnant (FLR) or procedural complications were reported in either group.

## 4. Discussion

The present study demonstrated a superiority in FLR volume increase of +48.10% at the first post-procedural assessment (12.4 ± 4 days) and +52.5% at the second postprocedural assessment (25.3 ± 4 days) for the LVD group when compared to the percentual increase achieved with PVE alone. This result is further confirmed by higher KGR/week rates for the LVD group which contextually showed a decreasing trend (KGR T_1_ 27.32 ± 16.86 vs. KGR T_2_ 17.19 ± 9.88) between different post-procedural time assessments, highlighting an earlier and faster hypertrophy induction for the LVD cohort. The theme of sufficient volume and function of FLR is especially pertinent in patients affected by perihilar cholangiocarcinoma (PHC) and scheduled for major hepatectomy [8]. PHLF stands as a key contributor to postoperative complications, and its occurrence is closely linked to both the adequacy of FLR volume and its functional capacity [20].

In PHC patients, the challenge posed by PHLF can be notably more demanding and severe compared to other surgical scenarios. This heightened severity arises from the synergistic impact of multiple adverse factors such as the aggressive nature of the surgical resection, necessitating extensive sacrifice of liver parenchyma, the presence of obstructive jaundice, which hampers liver regeneration mechanisms by inhibiting adenosine triphosphate production in mitochondria, and a diminished capacity of the liver for biosynthesis [21]. To this day, PVE is the hypertrophy technique of choice for preoperative optimization of FLR volume. This approach has demonstrated effective stimulation of lobar hypertrophy within a span of 2 to 4 weeks, resulting in a permissible FLR volume for surgical intervention [22]. Moreover, it has exhibited KGR rates exceeding 2.66% per week, showing proficient regenerative liver function while maintaining an excellent safety profile [23]. Nevertheless, PVE has two primary limitations, which become more significant in the context of PHC where surgical resection stands as the sole therapeutic recourse: nearly 20% of patients encounter insufficient attainment of hypertrophy volumes suitable for surgery, along with the possibility of disease progression during the interval between embolization and the surgical procedure [24]. Recently, to overcome the main drawbacks of PVE, two different hypertrophy approaches have been proposed. The ALPPS procedure demonstrated a significantly rapid induction of liver hypertrophy after the initial stage [11]; however, this strategy has been associated with high postprocedural morbidity and mortality rates. In the context of PHC, Olthof et al. compared outcomes between patients undergoing ALPPS and those undergoing standard resection with comparable FLR volumes. The ALPPS group exhibited a notably high mortality rate of 48% and a median survival of 6 months. Given these discouraging outcomes, the current role of ALPPS in PHC management is limited [25]. Conversely, LVD has demonstrated potential in fostering more substantial liver hypertrophy while maintaining a more tolerable incidence of complications. Initial series of LVD involving staggered embolization of the portal and hepatic venous systems failed to present a noteworthy temporal advantage over solitary PVE [26]. However, the concurrent embolization of both the portal and hepatic venous systems has demonstrated advantageous functional and volumetric results across various series. Guiu et al. conducted a comparison of changes in FLR volume and function between LVD and PVE in patients undergoing major hepatectomies [27]. The study concluded that LVD was linked to a more substantial and rapid enhancement in both liver hypertrophy and function. Nevertheless, all Klatskin’s tumor cases were excluded from the analysis. Similar results were found by Kobayashi et al. who also highlighted greater and more rapid FLR hypertrophy rates within the LVD subgroup when compared to PVE alone, irrespective of the diagnosis or the presence of underlying liver disease, for patients undergoing major hepatectomy [28]. Recently, the DRAGON collaborative group backed these results through a multicentric comparison between simultaneous LVD and PVE alone, concluding that LVD leads to increased resectability for various liver tumor types, induces higher liver hypertrophy resulting in larger FLR, and exhibits a safety profile comparable to PVE. The analysis reaffirms the advantage of LVD over PVE in terms of resectability and complication rates [14].

To the best of our knowledge, this is the first study that specifically compares functional and volumetric outcomes between PVE and LVD in patients affected by PHC and scheduled for preoperative optimization of FLR prior to major hepatectomy. Regarding volumetric outcomes, the results align with previous studies indicating a faster and higher induction of FLR hypertrophy [14,27,28]. LVD showed improved differential volumes at both post-procedural timelines (T_1_: 190.50 ± 116.50 vs. 112.28 ± 84.74; *p* = 0.034; T_2_: 216.38 ± 118.43 vs. 125.41 ± 93.45 *p* = 0.036) and greater sDH and KGR/week rates associated with a decreasing trend between T_1_ and T_2_ supporting the concept of a faster induction of liver regeneration. These results led to optimal post-procedural outcomes with a low rate of overall complications (just one case of segmental portal thrombosis effectively resolved with low molecular weight heparin), no evidence of oncologic progression, no drop-outs from the surgical program, with every patient undergoing resection within 29 days from the procedure. Despite these results not being statistically different from postprocedural outcomes of patients treated with PVE in our cohort, the role of LVD for PHC deserves further attention. The faster and greater induction of FLR hypertrophy by LVD could potentially address the main drawbacks of PVE (e.g., inadequate FLR volume and tumor progression during the waiting interval to surgery) which could lead to the exclusion of initially resectable patients from completing the surgical program or result in positive resection margin resections due to insufficient FLR volume. This advantage is crucial, especially in PHC, where surgical resection is the only therapeutic option [29].

One of the most significant findings of this study is related to the functional outcomes achieved with LVD. Not only was LVD superior to PVE in terms of improved FLR function (3.22 ± 0.55%/min/m^2^ vs. 2.62 ± 0.64%/min/m^2^; *p* = 0.041), but it also demonstrated a substantial increase in FLR function from the baseline at the first post-procedural functional evaluation (+67.7%, approximately three weeks after the procedure). In PHC, a large difference between volumetric and functional parameters is commonly observed during the preoperative evaluation of FLR. This difference can be attributed to the growth of hilar tumors, resulting in obstructive cholestasis, which negatively impacts optimal liver function [30]. To address this, preoperative biliary drainage placement is often performed. However, hemilobe biliary decompression has been associated with functional deterioration of the non-drained hemilobe while maintaining normal function in the decompressed hemilobe [31]. Additionally, early stages of intrahepatic cholestasis have been linked to a hypertrophic response, counterbalanced by a substantial decrease in hepatic function, which is frequently further diminished by drainage-related cholangitis. Given the challenging preoperative management of cholestasis-related damage in PHC patients, which increases the risk of insufficient postoperative FLR function to meet metabolic and biosynthetic needs, there is a renewed emphasis on proposing functional FLR assessment over volumetric evaluation [32]. To date, only a limited number of studies have focused on the role of PVE in enhancing FLR function. Those studies that evaluated functional assessment after PVE reported a functional increase of +51.9% at 3 weeks post-procedure, which is lower compared to our results (+67.7%) [33,34]. Furthermore, these studies were not specifically tailored to PHC patients, where there is an overall loss of liver function at baseline.

When comparing PVE to LVD, the DRAGON collaborative [14] and Kobayashi et al. [28] did not report functional outcomes, while Guiu et al. [27] assessed FLR function and reported a remarkable +68.2% increase after LVD, which was significantly higher compared to the functional gain in PVE (+29.8%). However, all patients affected by PHC were excluded from their analysis.

Finally, to further highlight the efficacy of this technique, no significant differences in complication rates between the two groups were found. Concerns had been raised regarding the potential compromise of hepatic outflow in LVD, theoretically increasing the risk of postprocedural complications such as bleeding, thrombosis, off-target embolization, or the exacerbation of intrahepatic stasis, potentially leading to an increased frequency of cholangitis episodes. However, none of these complications were encountered, highlighting the feasibility of the LVD technique.

### Limitations

Several limitations should be considered. Firstly, the retrospective study design could introduce inherent biases and confounding factors that may affect result validity. Secondly, the relatively small patient cohort assessed, particularly in light of the challenging preoperative management of patients with PHC, may potentially limit the generalizability of the findings. Liver venous deprivation as a hypertrophy technique is relatively recent in clinical practice. Moreover, its use in patients with PHC, who often present with obstructive jaundice, recurrent cholangitis, malnutrition, and inadequate liver remnant, is even more limited. These complexities and patient-specific challenges explain the low number of patients in our study. Thirdly, the varying intervals between embolization and postprocedural evaluations, along with potential delays in the subsequent surgical procedure due to the intricacies of patient care, could impact the interpretation of the results.

## 5. Conclusions

In conclusion, this study provides further confirmation of the faster and more substantial volumetric increase in FLR achievable after LVD, while maintaining a similar safety profile to PVE, even in the specific subset of patients affected by PHC. The quicker FLR volumetric gain suggests a potential enhancement in the waiting interval between the procedure and surgery, potentially reducing drop-out rates due to disease progression. Moreover, LVD demonstrated an exceptional improvement in FLR function, which could be the crucial factor in counterbalancing the poor baseline function caused by obstructive cholestasis and cholangitis-related complications in PHC. 

## Figures and Tables

**Figure 1 cancers-15-04363-f001:**
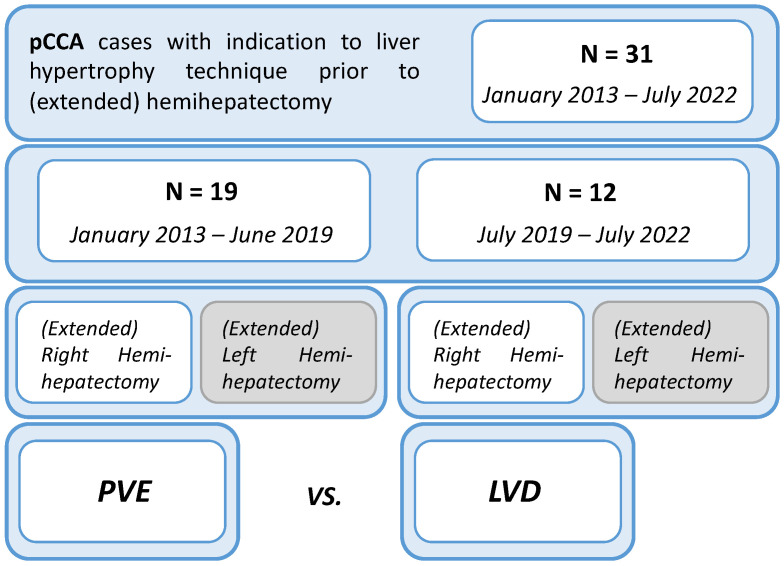
Study Design and Comparison of Hypertrophy Techniques for PHC Patients. Patients diagnosed with PHC and candidates for (extended) major hepatectomy were preoperatively treated using an institutional optimization protocol assessing FLR volume and function. Portal vein embolization (PVE) was the primary technique until June 2019, when associating liver partition and portal vein ligation (LVD) was introduced. The study compares the outcomes of the LVD cohort (July 2019–July 2022) with the PVE cohort (January 2013–June 2019) to evaluate the safety and efficacy of LVD.

**Figure 2 cancers-15-04363-f002:**
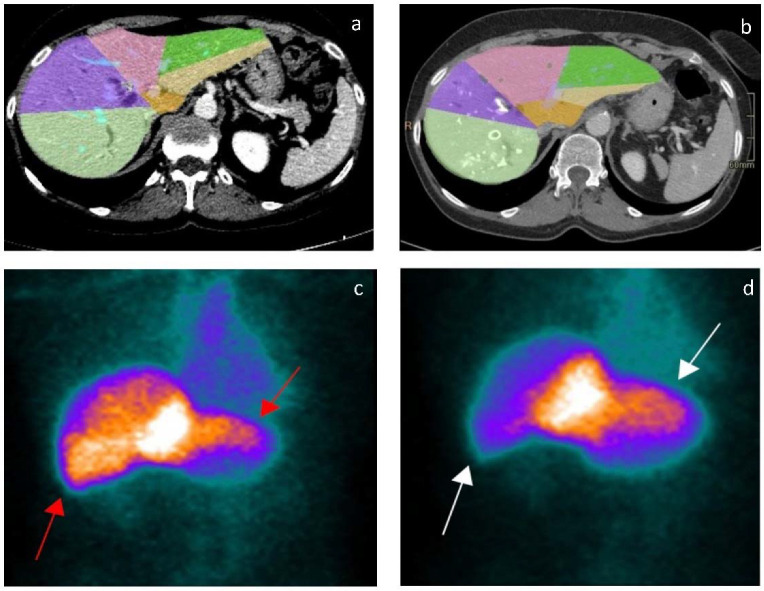
(**a**): Pre-procedural liver volumetric assessment obtained through Philips Intellispace Portal (Philips, Amsterdam, The Netherlands). (**b**): Post-procedural liver volumetric assessment showing left-lobe volumetric increase. (**c**): Pre-procedural (99m) Tc-Mebrofenin hepatobiliary scintigraphy showing a homogeneous captation between right and left lobe (red arrows). (**d**): Post-procedural (99m) Tc-Mebrofenin hepatobiliary scintigraphy showing a prevalent left-lobe captation (white arrows), underlining an increased metabolic activity compared to the right.

**Figure 3 cancers-15-04363-f003:**
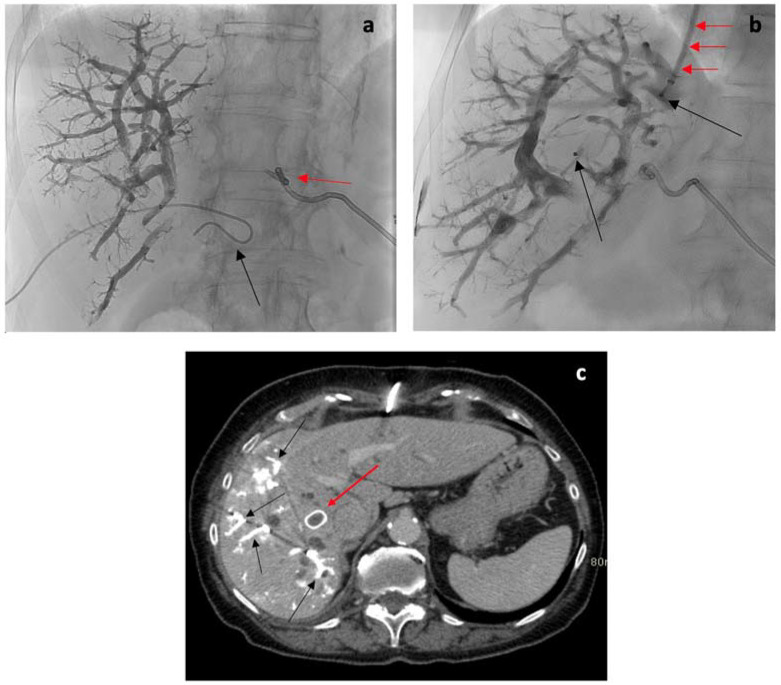
(**a**) Fluoroscopic image showing right portal branches filled with iperdense embolizing material (mixture of cyanoacrylate glue and lipiodol; 1:5), injected via a 4 Fr catheter (red arrow). Collaterally, external biliary drainage for the left hemisystem (black arrow). (**b**) Fluoroscopic image obtained after vascular plug deployment in the right hepatic vein through the transjugular approach. Black arrows: proximal and distal plug markers; red arrows: introducer sheath in the inferior cava/right hepatic vein. (**c**) Post-procedural CT axial scan. Red arrow: Amplatzer vascular plug located in the right hepatic; Black arrow: embolized right portal branches.

**Figure 4 cancers-15-04363-f004:**
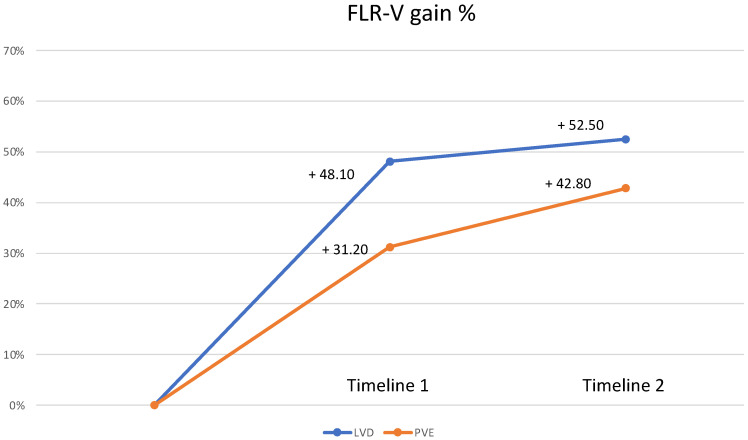
FLR-volume variation (mean with standard deviation) from baseline in the LVD (blue) and PVE (orange) cohorts according to the study timelines (*p* < 0.001).

**Figure 5 cancers-15-04363-f005:**
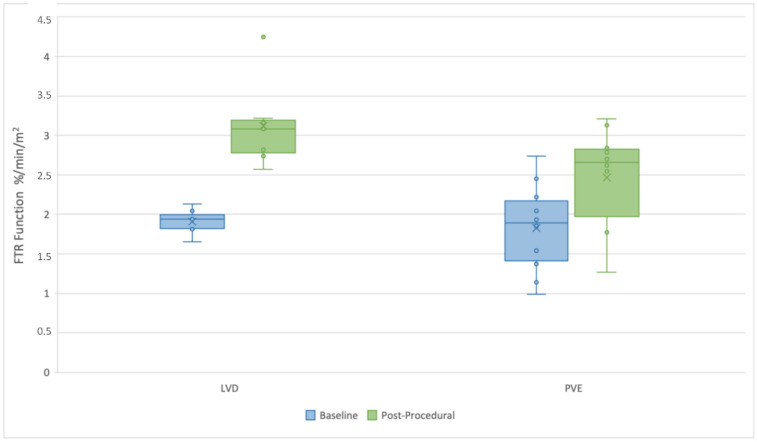
Rates of FTR function evaluated by (99m) Tc-Mebrofenin hepatobiliary scintigraphy at baseline and after inducing liver hypertrophy with either LVD or PVE. Preoperative LVD showed a significant improvement in FTR function (%/min/m^2^).

**Table 1 cancers-15-04363-t001:** Pre-procedural baseline characteristics.

	LVD (*n* = 12)	PVE (*n* = 19)	*p*-Value
Age, years	68.61 ± 8.06	66.53 ± 8.22	0.494
Sex, male	7 (58.3)	10 (52.6)	0.867
BSA (Kg/m^2^)	1.82 ± 0.19	1.83 ± 0.14	0.794
ASA ≥ 3	5 (41.7)	7 (36.8)	0.858
Total Bilirubin (mg/dL)	2.07 ± 2.78	1.27 ± 1.58	0.119
AST (U/L)	66.27 ± 30.62	74.53 ± 26.60	0.331
ALT (U/L)	88.81 ± 56.58	95.88 ± 37.27	0.423
Bismuth Type I II III IV	0 (0) 0(0) 11 (91.7) 1 (8.3)	0 (0) 0(0) 17 (89.5) 2 (10.5)	1.000
Preoperative biliary drainage	5 (41.7)	9 (47.4)	0.752
Preoperative cholangitis	3 (25)	4 (21.0)	0.823
Liver cirrhosis	0 (0)	1 (5.3)	0.806
sTLV	1524.11 ± 224.36	1510.62 ± 249.79	0.880
FLR (mL)	429.92 ± 164.98	464.73 ± 114.63	0.642
cFLR	27.71 ± 6.76	30.02 ± 6.53	0.346
sFLR	28.09 ± 9.54	27.86 ± 9.28	0.948
TLV function (%/min/m^2^)	4.99 ± 1.06	6.17 ± 2.25	0.237
FLR function (%/min/m^2^)	1.92 ± 0.14	1.87 ± 0.62	0.854

**Table 2 cancers-15-04363-t002:** (a) Post-procedural outcomes at first volumetric assessment (T_1_); (b) Post-procedural outcomes at second volumetric assessment (T_2_); (c) Post-procedural functional outcomes.

	LVD (*n* = 12)	PVE (*n* = 19)	*p*-Value
	Post-procedural timeline 1 (12.4 ± 4 days)		a
FLR (mL)	628.84 ± 229.60	577.33 ± 187.18	0.601
PostFLR–PreFLR (mL)	190.50 ± 116.50	112.28 ± 84.74	0.034
cFLR (%)	36.83 ± 8.70	31.82 ± 7.32	0.329
sFLR (%)	41.22 ± 12.76	37.58 ± 8.44	0.530
sDH (%)	48.45 ± 26.87	39.67 ± 22.38	0.006
KGR/week (%)	27.32 ± 16.86	15.71 ± 9.82	< 0.001
	Post-procedural timeline 2 (25.3 ± 4)		b
FLR (mL)	639.75 ± 247.64	623.21 ± 192.36	0.725
PostFLR–PreFLR (mL)	216.38 ± 118.43	125.41 ± 93.45	0.036
cFLR (%)	38.42 ± 8.67	34.21 ± 7.43	0.368
sFLR (%)	41.65 ± 14.09	39.41 ± 9.21	0.563
sDH (%)	55.60 ± 31.02	42.29 ± 21.09	0.003
KGR/week (%)	17.19 ± 9.88	9.89 ± 14.62	0.034
	Functional Analysis		c
Availability of pre- and post-procedural scintigraphy	10 (83.3)	16 (84.2)	0.806
Post-procedural FLR function (%/min/m^2^)	3.22 ± 0.55	2.62 ± 0.64	0.041

**Table 3 cancers-15-04363-t003:** Post-procedural outcomes referencing technical success, complications, and surgical drop-out.

	LVD (*n* = 12)	PVE (*n* = 19)	*p*-Value
Technical Success	12 (100)	19 (100)	1.000
Complications	1 (8.3)	1 (5.3)	0.632
Surgery	12 (100)	17 (89.47)	0.509
Time to surgery (days)	29 ± 3	35 ± 4	0.368
PHLF, ISGLS B/C	0 (0)	1 (5.26)	0.613
Surgery Drop-out			
Oncological progression	0 (0)	2 (10.5)	0.509
Insufficient FLR	0 (0)	0 (0)	1.000
Procedural complications	0 (0)	0 (0)	1.000

## Data Availability

The data that support the findings of this study are available from the corresponding author, LA, upon reasonable request.

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
