# Peer review of "Comparing Liver Venous Deprivation and Portal Vein Embolization for Perihilar Cholangiocarcinoma: Is It Time to Shift the Focus to Hepatic Functional Reserve Rather than Hypertrophy?"

_cancers, 2023, doi:10.3390/cancers15174363_

Round 1
Reviewer 1 Report
The authors intended to compare liver venous deprivation with portal vein embolization to induce hypertrophy in patients with perihilar cholangiocarcinoma.
The study addresses an interesting and important subject. Nonetheless, I have several concerns that should be answered by the authors.
The total number of patients is quite low. In particular, in light of the heterogeneity of liver parenchyma in patients.
The preoperative information on liver function did not result in an improved clinical outcome. Is Tc-Mebrofenin adequately refelcting liver function?
Author Response
Dear Editor,
We are pleased to resubmit to “Cancers” the modified version of our article entitled: “Comparing Liver Venous Deprivation and Portal Vein Embolization for Perihilar Cholangiocarcinoma: is it time to shift the focus to hepatic functional reserve rather than hypertrophy?”.
We appreciate the efforts of the Editor and Reviewers in reviewing our manuscript and we are grateful for the opportunity to review and resubmit our paper.
We have reexamined the manuscript based on your valuable comments.
A point-by-point response to the Editors’ and Reviewers’ comments can be found below. We believe that the edits made have strengthened our paper.
Reviewer #1
1)The authors intended to compare liver venous deprivation with portal vein embolization to induce hypertrophy in patients with perihilar cholangiocarcinoma. The study addresses an interesting and important subject. Nonetheless, I have several concerns that should be answered by the authors. The total number of patients is quite low. In particular, in light of the heterogeneity of liver parenchyma in patients.
We appreciate your thoughtful comments and concerns regarding our manuscript entitled. We value your feedback and would like to address your points directly:
We acknowledge the limitation of the total number of patients included in our study. We have taken your feedback into account and have further highlighted this limitation in the "Limitations" section of the manuscript. The text now reads as follows:
“Several limitations should be considered. Firstly, the retrospective study design could introduce inherent biases and confounding factors that may affect result validity. Secondly, the relatively small patient cohort assessed, particularly in light of the challenging preoperative management of patients with PHC, may potentially limit the generalizability of the findings. Liver venous deprivation as a hypertrophy technique is relatively recent in clinical practice. Moreover, its use in patients with PHC, who often present with obstructive jaundice, recurrent cholangitis, malnutrition, and inadequate liver remnant, is even more limited. These complexities and patient-specific challenges explain the low number of patients in our study. Thirdly, the varying intervals between embolization and postprocedural evaluations, along with potential delays in the subsequent surgical procedure due to the intricacies of patient care, could impact the interpretation of the results.”
Your concern about the limited number of patients is valid, and we would like to provide additional context. Liver venous deprivation as a hypertrophy technique is a relatively recent development in clinical practice, with its application beginning around 2016. Moreover, its use in patients with perihilar cholangiocarcinoma, a particularly challenging and rare condition, is even more limited. This scarcity of cases, coupled with the preexisting difficulties these patients face, such as obstructive jaundice, recurrent cholangitis, malnutrition, and inadequate liver remnant, explains the low number of patients in our study. These factors collectively underscore the challenges inherent in conducting research in this patient cohort and further justify the limited sample size.
2)The preoperative information on liver function did not result in an improved clinical outcome. Is Tc-Mebrofenin adequately reflecting liver function?
We appreciate your valuable comment regarding the preoperative assessment of liver function and its correlation with clinical outcomes. Your observation raises an important aspect of our study.
In our current study, we acknowledge that the number of patients included is limited (See Response 1 Reviewer 1). Consequently, the statistical power to definitively establish a relationship between preoperative liver function, as reflected by Tc-Mebrofenin, and clinical outcomes is constrained. We completely agree that a larger patient cohort would provide more robust insights into this relationship.
However, it is worth noting some trends in our data that offer intriguing insights. Notably, our results indicated that there were no cases of post-hepatectomy liver failure (PHLF, ISGLS B/C) in the Liver Venous Deprivation (LVD) group, while one case (5.26%) occurred in the Portal Vein Embolization (PVE) group (p = 0.613). Furthermore, two (10.5%) patients in the PVE group had to drop out due to oncological progression, whereas no drop-outs occurred in the LVD group (p = 0.509). While these results did not reach statistical significance in our limited cohort, they do suggest a noteworthy trend.
Considering these trends, it is reasonable to speculate that a larger and more diverse cohort of patients might provide the statistical power needed to establish the potential benefits of LVD more definitively. The absence of PHLF and fewer drop-outs in the LVD group is encouraging and hints at the possibility of improved clinical outcomes. However, as you rightly pointed out, this would require a larger sample size to confirm conclusively.
It is indeed noteworthy that several liver remnant function tests are currently available in clinical practice. However, none of these tests have been universally adopted as a clinical standard to date. For instance, Japanese institutions often prefer the use of tests like ICG (e.g., ICGR15 - indocyanine green retention rate at 15 minutes or ICGK-F - indocyanine green clearance of the future liver remnant) to evaluate liver function. These tests offer valuable insights into liver function, particularly in the context of planned hepatic resections.
Nonetheless, it is important to acknowledge that these tests, while effective in many cases, may not be suitable for all patients. For example, individuals with constitutional ICG excretory defects or those prone to allergic reactions may not benefit from ICG-based assessments. In such cases, alternative approaches like TC-mebrofenin scintigraphy or 99mTc-GSA scintigraphy provide valid and valuable alternatives to evaluate liver function.
We would like to express our sincere gratitude to the editors and the reviewers for the precise feedback and extremely helpful comments which we believe were essential in improving the overall manuscript’s quality.
This manuscript has not been published and is not under consideration for publication elsewhere. We have no conflicts of interest to disclose.
Thanks in advance for the attention and the valuable time you will spend for us.
On behalf of all the Authors,
Prof. Luca Aldrighetti, MD, PhD
Head, Department of Hepatobiliary Surgery Division, IRCCS San Raffaele Hospital
Via Olgettina 58-60, 20132, Milan, Italy

Reviewer 2 Report
Congratulations, your manuscript is very interesting. I would like to make the following suggestions:
Line 47: "Due to the standardized of surgical approach...". Please review, if I may suggest "Due to standardized surgical approach..."
Line 48: "...mandatorily involving (extended) right or left hepatectomies...". Extended resections are not mandatory in perihilar cholangiocarcinoma Bismuth type I, IIa or IIb. This study includes only Bismuth type III and IV in which extended resections are indeed mandatory. Please rephrase and clarify.
Line 51: "extensive parenchymal demolitions". Please review, demolition is not a commonly used surgical term, if I may suggest: "extensive parenchymal resections".
Line 67: "over 20% of treated patients are necessarily excluded from surgical resection". Please review, the patients are excluded only when absolute contraindications are observed, therefore, no patient is unnecessarily excluded.
Line 80 and 81: "but a specific evaluation...but deserves specific attention..." Please review, two consecutives phrases begining with adversative connectors may create confusion in the reader.
Line 121 and 122: "...in candidates to hypertrophy techniques. eventually, a biliary stenting..." Please review, either 'eventually' should have a capital 'E' or the phrase should be modified.
Line 146: "(Figure 2a,b). Hepatobiliary scintigraphy..." Please review, the paragraph should not start with parenthesis.
Line 148: "...the total liver function was multiplied by the FLRF..." Please review, this is not the formula used in Reference #19. In that reference, FLRF is divided by total liver funcion in order to obtain a percentage.
Line 204: Please define 'PPSn'
Line 212: Given the sample size, continuous variables should be expressed with median and interquartile range (IQR).
Line 214: Mann-Whitney U test for paired samples could be used in this study to analyze the difference between the LVD group before and after intervention, meaning ONE group in TWO different points of time. However, the Mann-Whitney U test is used in the manuscript to compared TWO groups in ONE point of time, therefore, Mann-Whitney U test for independent samples should be included in the 'Methods' section.
Line 252. The use of letters within a Table is not common in medical literature, however, I see the value of its application. I believe it would be clearer for the reader is placed BEFORE the subheading, i.e. 'a. Post-procedural timeline 1' instead of AFTER the subheading.
Line 293. The table lines are duplicated in my PDF version of the manuscript.
Line 298. "...using curative doses of low molecular weight heparin". Please review, the most common medical term is "therapeutic" dosing as opposed to "prophylactic" dosing.
Line 307. As stated in the 'Introduction' and 'Discussion' section of this manuscript, the morbidity of liver resection for perihilar cholangiocarcinoma is very high. The message to the reader is unclear when stated that only one patient developed a postoperative complication (grade B/C PHLF). To try and list all complications would be unpractical, but I would suggest to include in the 'Methods' section the "Comprehensive Complication Index" (CCI) and present the difference in median (IQR) CCI between the groups.
Line 423. I believe that a fourth limitation of this study is the lack of clinical impact after resection with the currently available data in the manuscript, This could change either after including CCI or a larger sample size.
The manuscript benefit from minor english language review.
Author Response
Dear Editor,
We are pleased to resubmit to “Cancers” the modified version of our article entitled: “Comparing Liver Venous Deprivation and Portal Vein Embolization for Perihilar Cholangiocarcinoma: is it time to shift the focus to hepatic functional reserve rather than hypertrophy?”.
We appreciate the efforts of the Editor and Reviewers in reviewing our manuscript and we are grateful for the opportunity to review and resubmit our paper.
We have reexamined the manuscript based on your valuable comments.
A point-by-point response to the Editors’ and Reviewers’ comments can be found below. We believe that the edits made have strengthened our paper.
Reviewer #2
1)Line 47: "Due to the standardized of surgical approach...". Please review, if I may suggest "Due to standardized surgical approach..."
Thank you for your helpful comment regarding the manuscript. We appreciate your careful review, and we have made the suggested change to improve the clarity of the sentence. The text now reads as follows: “Due to standardized surgical approach for hepatic hilum tumors, which mandates (extended) right or left hepatectomies in conjunction with bile duct and caudate lobe resection in Bismuth III or IV cases, postoperative morbidity and mortality rates rank among the highest within the HPB field.”
2)Line 48: "...mandatorily involving (extended) right or left hepatectomies...". Extended resections are not mandatory in perihilar cholangiocarcinoma Bismuth type I, IIa or IIb. This study includes only Bismuth type III and IV in which extended resections are indeed mandatory. Please rephrase and clarify.
Thank you again for your insight. The text now reads as follows: “Due to standardized surgical approach for hepatic hilum tumors, which mandates (extended) right or left hepatectomies in conjunction with bile duct and caudate lobe resection in Bismuth III or IV cases, postoperative morbidity and mortality rates rank among the highest within the HPB field.”
3)Line 51: "extensive parenchymal demolitions". Please review, demolition is not a commonly used surgical term, if I may suggest: "extensive parenchymal resections".
Thank you for your valuable feedback. We have made the suggested change to enhance the clarity of the text. Line 51 now reads: "extensive parenchymal resections."
4)Line 67: "over 20% of treated patients are necessarily excluded from surgical resection". Please review, the patients are excluded only when absolute contraindications are observed, therefore, no patient is unnecessarily excluded.
We appreciate your careful consideration of our manuscript and your insightful comment regarding the exclusion of patients from surgical resection following PVE.
You are absolutely correct, and we apologize for any confusion in our previous statement. Patients are indeed excluded from surgical resection only when absolute contraindications are observed, ensuring that no patient is unnecessarily excluded from this potentially life-saving procedure. We have revised the sentence accordingly to accurately reflect this important point. The text now reads as follows: “Although PVE has generally exhibited effective contralateral hypertrophy induction within 4-6 weeks, approximately 20% of treated patients are excluded from surgical resection due to the emergence of absolute contraindications, such as inadequate liver regeneration or tumor progression.”
5)Line 80 and 81: "but a specific evaluation...but deserves specific attention..." Please review, two consecutives phrases begining with adversative connectors may create confusion in the reader.
Thank you for noticing. Line 80 and 81 now read as follows: “These favorable results have been documented in cases of secondary liver tumors and HCC. Nevertheless, there is currently a gap in our understanding of the application of LVD in the context of pCCA. This area merits particular attention, especially considering the high-risk nature of these patients, both in terms of the perioperative and oncological outcomes.”
6)Line 121 and 122: "...in candidates to hypertrophy techniques. eventually, a biliary stenting..." Please review, either 'eventually' should have a capital 'E' or the phrase should be modified.
Thank you for your keen observation. We have capitalized 'Eventually' in the sentence as per your suggestion to ensure proper grammar and readability. Line 121 and 122 now read as follow: “Eventually, a biliary stenting of right bile ducti, via endoscopic retrograde cholangiography, was indicated based on the total bilirubin level following PTBD.”
7)Line 146: "(Figure 2a,b). Hepatobiliary scintigraphy..." Please review, the paragraph should not start with parenthesis.
Thank you for your comment. The sentence was revised accordingly to ensure proper formatting. The text now reads as follows: Hepatobiliary scintigraphy (HBS) was conducted before the procedure, following established protocols.
8) Line 148: "...the total liver function was multiplied by the FLRF..." Please review, this is not the formula used in Reference #19. In that reference, FLRF is divided by total liver funcion in order to obtain a percentage.
We greatly appreciate your attention to detail, and we have revised the sentence in question to enhance clarity and align it with the definition provided in Reference #19. The sentence now accurately reflects that FLRF is divided by total liver function to obtain a percentage. Line 148 now reads as follows: “The calculation of the future liver remnant function (FLRF) involved delineating the counts within the FLR, dividing this by the total liver counts, and then multiplying this factor with the total liver 99mTc-mebrofenin uptake. The result was expressed as a percentage per minute per square meter (%/min/m²).”
9) Line 204: Please define 'PPSn'
Apologies for the confusion. The formula has been changed as follows: The calculation for FLR increase was as follows: FLR increase = (FLRpost-procedural – FLRbaseline) x 100%
10) Line 212: Given the sample size, continuous variables should be expressed with median and interquartile range (IQR).
We appreciate the reviewer's suggestion to consider using median and interquartile range (IQR) for the presentation of continuous variables in our study. However, we chose to report these variables as mean and standard deviation for several reasons. Firstly, the data distribution for the reported continuous variables approximates normality, displaying a symmetric pattern that aligns with the assumptions of mean and standard deviation. This choice enhances the interpretability of our findings, as it is a more familiar method for clinicians and researchers accustomed to these statistics. Despite the sample size limitation, we believe that for our sample, mean and standard deviation remain informative measures for conveying central tendency and variability. Moreover, we aim to facilitate comparisons with other studies or reference data, where mean and standard deviation are commonly employed. In our context, these statistics hold specific clinical relevance and accurately capture the data characteristics we intend to convey.
11) Line 214: Mann-Whitney U test for paired samples could be used in this study to analyze the difference between the LVD group before and after intervention, meaning ONE group in TWO different points of time. However, the Mann-Whitney U test is used in the manuscript to compared TWO groups in ONE point of time, therefore, Mann-Whitney U test for independent samples should be included in the 'Methods' section.
Thank you once again for your precise feedback. We sincerely apologize for the lack of clarity in the statistical analysis paragraph. We have since revised the paragraph to enhance its clarity and comprehensibility. The “Statistical Analyses” paragraph now reads as follows: “The statistical analyses were performed with IBM SPSS Statistics 28 (IBM Corp., Armonk, NY; USA). The Shapiro-Wilk normality test was conducted to evaluate the normality of the distribution. Continuous variables exhibiting normal distribution were presented as means ± standard deviation, while those with non-normal distribution were expressed as medians with their respective ranges. To analyze continuous variables, the Student's t-test was employed for normally distributed data, and the Mann-Whitney U test for independent samples was used for non-normally distributed data. The data was subjected to descriptive assessment and frequencies employed for categorical or ordinal variables. Qualitative variables were assessed using either the c2 test or Fisher’s exact test, as deemed suitable. A significance level of p < 0.05 was adopted to determine statistical significance.”
12) Line 252. The use of letters within a Table is not common in medical literature, however, I see the value of its application. I believe it would be clearer for the reader is placed BEFORE the subheading, i.e. 'a. Post-procedural timeline 1' instead of AFTER the subheading.
We appreciate your thoughtful feedback on the formatting of our table. Your suggestion to place letters before the subheading to improve clarity is well-received. We have made the necessary adjustments to Table 2 accordingly.
13) Line 293. The table lines are duplicated in my PDF version of the manuscript.
We sincerely apologize for any inconvenience caused by the duplicated table lines in your PDF version of the manuscript. This issue may be attributed to a formatting error during the document conversion process. We will investigate and rectify this problem promptly to ensure the final version of the manuscript is error-free.
14) Line 298. "...using curative doses of low molecular weight heparin". Please review, the most common medical term is "therapeutic" dosing as opposed to "prophylactic" dosing.
Thank you for pointing out the medical terminology regarding dosing. We have made the appropriate change to use "therapeutic" instead of "curative" in the manuscript to accurately reflect the common medical terminology. Line 298 now reads as follows: “However, both cases were successfully treated conservatively using therapeutic doses of low molecular weight heparin (grade-2).”
15) Line 307. As stated in the 'Introduction' and 'Discussion' section of this manuscript, the morbidity of liver resection for perihilar cholangiocarcinoma is very high. The message to the reader is unclear when stated that only one patient developed a postoperative complication (grade B/C PHLF). To try and list all complications would be unpractical, but I would suggest to include in the 'Methods' section the "Comprehensive Complication Index" (CCI) and present the difference in median (IQR) CCI between the groups.
Thank you for your valuable suggestions. We would like to clarify that the primary aim of our study was not to assess overall postoperative complications in patients undergoing surgery for perihilar cholangiocarcinoma. Instead, our focus was on evaluating postprocedural complications and specifically identifying cases of grade B/C Post-Hepatectomy Liver Failure (PHLF). This specific focus allowed us to investigate the impact of the LVD on liver function and postprocedural outcomes, which aligns with the research objectives outlined in the 'Introduction' and 'Methods' sections of our manuscript.
Two patients, one from each group, experienced post-procedural complications. In both cases, these complications manifested as segmental portal thrombosis in segment 2, as observed at first post-procedural assessment (T1). However, both cases were successfully treated conservatively using therapeutic doses of low molecular weight heparin (grade-2). No additional procedural complications, such as bilomas, hepatic bleeding, or arteriovenous fistulas (AVF), were observed. Therefore, for postprocedural complications, calculating the Comprehensive Complication Index (CCI) would not add any additional information or reveal statistical differences. Regarding postoperative PHLF grade B/C, it is noteworthy that there were no cases of post-hepatectomy liver failure (PHLF, ISGLS B/C) in the LVD group, while one case (5.26%) occurred in the PVE group (p = 0.613). Furthermore, in this context, calculating the Comprehensive Complication Index (CCI) would not have added any additional information or revealed statistical differences.
16) Line 423. I believe that a fourth limitation of this study is the lack of clinical impact after resection with the currently available data in the manuscript, This could change either after including CCI or a larger sample size.
Thank you for your valuable input regarding potential limitations in our study. We appreciate your thoughtful consideration.
We acknowledge the limitation of the total number of patients included in our study. We have taken your feedback into account and have further highlighted this limitation in the "Limitations" section of the manuscript. The text now reads as follows:
“Several limitations should be considered. Firstly, the retrospective study design could introduce inherent biases and confounding factors that may affect result validity. Secondly, the relatively small patient cohort assessed, particularly in light of the challenging preoperative management of patients with PHC, may potentially limit the generalizability of the findings. Liver venous deprivation as a hypertrophy technique is relatively recent in clinical practice. Moreover, its use in patients with PHC, who often present with obstructive jaundice, recurrent cholangitis, malnutrition, and inadequate liver remnant, is even more limited. These complexities and patient-specific challenges explain the low number of patients in our study. Thirdly, the varying intervals between embolization and postprocedural evaluations, along with potential delays in the subsequent surgical procedure due to the intricacies of patient care, could impact the interpretation of the results.”
Your concern about the limited number of patients is valid, and we would like to provide additional context. Liver venous deprivation as a hypertrophy technique is a relatively recent development in clinical practice, with its application beginning around 2016. Moreover, its use in patients with perihilar cholangiocarcinoma, a particularly challenging and rare condition, is even more limited. This scarcity of cases, coupled with the preexisting difficulties these patients face, such as obstructive jaundice, recurrent cholangitis, malnutrition, and inadequate liver remnant, explains the low number of patients in our study. These factors collectively underscore the challenges inherent in conducting research in this patient cohort and further justify the limited sample size. We believe that further investigation with larger samples and prospective studies would eventually highlight the clinical impact of LVD specifically for pCCA.
We would like to express our sincere gratitude to the editors and the reviewers for the precise feedback and extremely helpful comments which we believe were essential in improving the overall manuscript’s quality.
This manuscript has not been published and is not under consideration for publication elsewhere. We have no conflicts of interest to disclose.
Thanks in advance for the attention and the valuable time you will spend for us.
On behalf of all the Authors,
Prof. Luca Aldrighetti, MD, PhD
Head, Department of Hepatobiliary Surgery Division, IRCCS San Raffaele Hospital
Via Olgettina 58-60, 20132, Milan, Italy
